# Serological identification of SARS-CoV-2 infections among children visiting a hospital during the initial Seattle outbreak

Adam S. Dingens [1], Katharine H. D. Crawford[1,2,3], Amanda Adler[4], Sarah L. Steele[4], Kirsten Lacombe[4], Rachel Eguia[1], Fatima Amanat [5,6], Alexandra C. Walls[7], Caitlin R. Wolf[8], Michael Murphy[9], Deleah Pettie [9], Lauren Carter[9], Xuan Qin[4], Neil P. King[7,9], David Veesler [7], Florian Krammer [6], Jane A. Dickerson[4,10], Helen Y. Chu[8], Janet A. Englund[4,11✉] & Jesse D. Bloom [1,2,12✉]

Children are strikingly underrepresented in COVID-19 case counts. In the United States, children represent 22% of the population but only 1.7% of confirmed SARS-CoV-2 cases as of April 2, 2020. One possibility is that symptom-based viral testing is less likely to identify infected children, since they often experience milder disease than adults. Here, to better assess the frequency of pediatric SARS-CoV-2 infection, we serologically screen 1,775 residual samples from Seattle Children's Hospital collected from 1,076 children seeking medical care during March and April of 2020. Only one child was seropositive in March, but seven were seropositive in April for a period seroprevalence of ≈1%. Most seropositive children (6/8) were not suspected of having had COVID-19. The sera of seropositive children have neutralizing activity, including one that neutralized at a dilution > 1:18,000. Therefore, an increasing number of children seeking medical care were infected by SARS-CoV-2 during the early Seattle outbreak despite few positive viral tests.

[1] Basic Sciences and Computational Biology, Fred Hutchinson Cancer Research Center, Seattle, WA 98109, USA. [2] Department of Genome Sciences, University of Washington, Seattle, WA 98195, USA. [3] Medical Scientist Training Program, University of Washington, Seattle, WA 98195, USA. [4] Division of Infectious Disease, Seattle Children's Hospital, Seattle, WA 98105, USA. [5] Graduate School of Biomedical Sciences, Icahn School of Medicine at Mount Sinai, New York, NY 10029, USA. [6] Department of Microbiology, Icahn School of Medicine at Mount Sinai, New York, NY 10029, USA. [7] Department of Biochemistry, University of Washington, Seattle, WA 98195, USA. [8] Division of Allergy and Infectious Diseases, University of Washington, Seattle, WA 98195, USA. [9] Institute for Protein Design, University of Washington, Seattle, WA 98195, USA. [10] Department of Laboratory Medicine and Pathology, University of Washington, Seattle, WA 98195, USA. [11] Department of Pediatrics, University of Washington, Seattle, WA 98195, USA. [12] Howard Hughes Medical Institute, Seattle, WA 98103, USA. ✉email: janet.englund@seattlechildrens.org; jbloom@fredhutch.org

One of the first cases of community transmission of SARS-CoV-2 in the United States was identified in the greater Seattle area in late February, 2020[1,2]. By late March, thousands of cases had been identified in Washington state by viral reverse transcription-polymerase chain reaction (RT-PCR) testing, mostly among adults (https://www.doh.wa.gov/Emergencies/Coronavirus). Schools closed statewide on March 17, and a statewide stay-at-home order was issued the next week. March and April of 2020 are therefore critical months for understanding the early dynamics of the SARS-CoV-2 pandemic in the Seattle area.

Children are underrepresented in confirmed SARS-CoV-2 case counts[3–5]. Since SARS-CoV-2-infected children often experience little or no disease[3,6–9], one possibility is that pediatric infections are oftentimes missed by symptom-based administration of viral tests. Therefore, we sought to identify infections using an approach independent of symptom-based viral testing. Serological assays, which detect antibodies induced by infection, provide such an approach. When interpreting these assays in a temporal context, note that individuals do not become seropositive until ≈1–2 weeks post symptom onset[10–14], while PCR-based testing generally only detects viral RNA during the first few weeks after symptom onset[11,12].

## Results and discussion

### Serological screening to residual pediatric sera samples.
We serologically screened 1775 residual serum samples from Seattle Children's Hospital that were collected between March 3 and April 24, 2020 following approval from the Human Subjects Institutional Review Board. These samples were collected from 1076 unique children who visited the hospital and received blood draws for any reason, including respiratory illnesses, surgery, or ongoing medical care. Demographics and the reason for medical admission are presented below with the results of our serological testing. The generalizability of this study population to all children in Seattle is unknown, particularly because hospital visitors were primarily those with urgent medical needs during the statewide stay-at-home order.

We used a multi-assay serological testing approach based on an enzyme-linked immunosorbent assay (ELISA) protocol that recently received emergency use authorization from New York State and the FDA[15,16], although we increased stringency by adding a second-validation ELISA and confirming putative seropositives with the Abbott SARS-CoV-2 IgG chemiluminescent microparticle immunoassay (CMIA), which identifies IgG antibodies to the nucleocapsid protein, and has been shown to have 99.9% specificity and 100% sensitivity for samples taken greater than 17 days post symptom onset[17]. Furthermore, as described below, we confirmed that seropositive samples had activity in pseudovirus-neutralization assays.

We first screened all sera at a 1:50 dilution in an ELISA for IgG binding to the SARS-CoV-2 spike receptor-binding domain (RBD) and compared the results to a negative control consisting of a pool of sera collected in 2017 and 2018 (Fig. 1a). We identified 102 of 1775 samples with readings that exceeded the average of the negative controls by >5 standard deviations. These preliminary hits were further assessed with IgG ELISAs using serial dilutions of sera against two antigens: RBD and pre-fusion-stabilized spike ectodomain trimer (Fig. 1b). As negative controls, we included twelve serum samples and two serum pools collected before 2020; we also tested some pediatric samples that were negative in the initial RBD screen. We summarized the ELISAs by calculating the area under the curve (AUC), and called samples as putatively seropositive if the AUC exceeded the average of the negative controls by >5 standard deviations for both RBD and

spike (Fig. 1b). We then performed a final validation by testing with the Abbott CMIA all putative seropositives from the ELISAs, as well as most other samples with sufficient volume that passed the initial RBD screen. The AUCs for the RBD and spike ELISAs were highly correlated with each other and the Abbott CMIA index values (Fig. 1c).

Visual inspection of the serological assay results in Fig. 1b provides a sense of the trade-offs in calling seropositivity. Our assays included controls from adults with RT-PCR-confirmed infections, as well as samples from five children with confirmed infections collected at various times post symptom onset, including children from outside the study population (see "Methods—Study participants" for additional details). Neither adult nor child samples <1 week post symptom onset were seropositive by our criteria, consistent with prior reports that infected individuals generally do not become seropositive until at least a week post symptom onset[10–14]. However, all samples ≥1 week post symptom onset were seropositive, and in most cases, the signal greatly exceeded pre-2020 negative controls. Samples from children who never tested positive for SARS-CoV-2 are shown at the far right in Fig. 1b. The samples in this set that we classified as seropositive had readings comparable to confirmed infections ≥1 week post symptom onset. However, some samples from children who had not tested positive for the virus had readings that exceeded pre-2020 negative controls, but were weaker than confirmed-infection samples ≥1 week post symptom onset, and below the cutoffs in one or more assay. Our stringent criteria classify these samples as seronegative, although it is possible that some represent recent infections in children who had not yet developed robust antibody responses. Indeed, one symptomatic child who tested positive by RT-PCR had multiple samples taken <1 week post symptom onset that were seronegative in all assays (Fig. 1b, "child < 1 week"). Of note, two samples passed the ELISA cutoffs but were negative in Abbott CMIA and therefore classified as seronegative—but no samples were positive in the Abbott CMIA and negative in the ELISAs. Given that the Abbot CMIA has been shown to have 100% sensitivity for samples greater than 17 days post symptom onset[17], we suspect that any infections missed by our serological screening must have been quite recent (such as the <1-week RT-PCR-confirmed infections in Fig. 1b, and possibly the two ELISA-positive but Abbot-negative samples).

Overall, our assays identified ten seropositive samples from eight different children in the study population. These include seven seropositive samples from six children who had never tested positive for SARS-CoV-2 (far-right facet of Fig. 1b), as well as the samples from the children labeled as 2 and 3 weeks post symptom onset in Fig. 1b. The seropositive samples from the children 1 and 4 weeks post symptom onset are from RT-PCR-confirmed SARS-CoV-2 infections that were referred to Seattle Children's Hospital, and are not part of the residual serum pool that makes up our study population (see "Methods" for details).

### Timing and clinical information of seropositive samples.
We next examined the frequency of seropositive samples in the context of the temporal dynamics of the SARS-CoV-2 outbreak in Seattle (Fig. 2a). The first seropositive sample was collected in late March, and there were no additional seropositive samples until the second week of April. From that time on, a low but steady fraction of samples were seropositive for a period seroprevalence in our study population of ≈1% in April (Fig. 2a, b). While this convenience sample from children seeking medical care is unlikely to be representative of all children in Seattle, the period seroprevalence measurements for our population are similar to all-age cumulative incidence estimates for the Seattle region based on viral testing and mortality data[20,21], given the ≈1–2-week lag

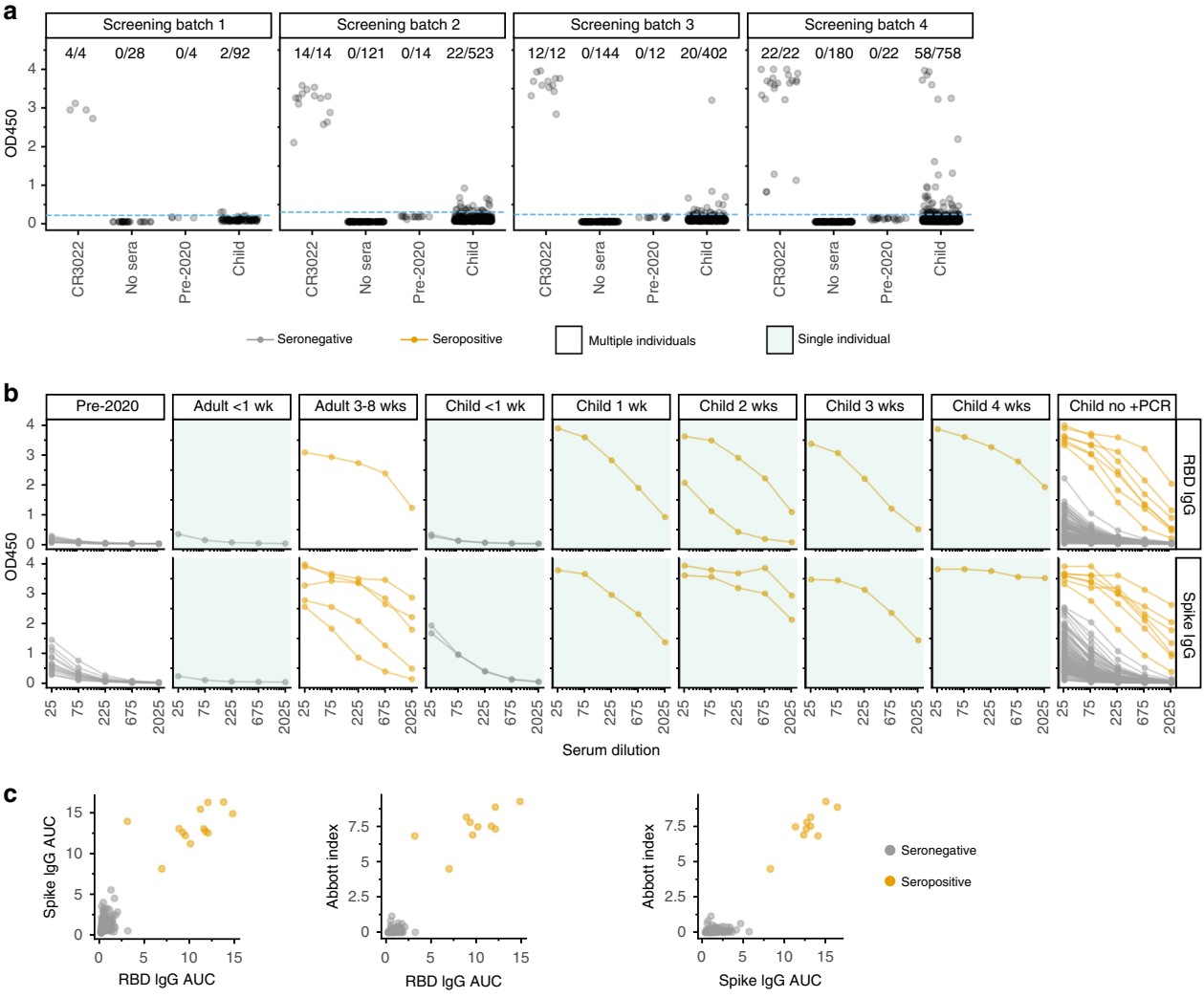

**Fig. 1 Multistep serological testing. a** We screened 1775 child samples by ELISA to RBD at a single dilution in four batches, with CR3022 antibody[18,19] as a positive control and pre-2020 sera as a negative control. Samples with OD450 readings that exceeded pre-2020 sera by >5 standard deviations (dotted blue line) were considered potential hits. **b** All potential hits from the initial screen, as well as some screen-negative samples and additional controls, were tested at serial dilutions for binding to RBD (top) or full spike (bottom). Samples from adults or children with RT-PCR-confirmed infections are labeled by weeks post symptom onset; all the remaining samples from children with no positive RT-PCR test are in the rightmost facet. Samples were classified as seropositive (orange) if the AUC exceeded pre-2020 negative controls by >5 standard deviations in both ELISA assays, and they were positive in the Abbott CMIA. All samples in green-shaded panels were from the same individual, whereas unshaded panels show samples from multiple individuals; see "Methods" for more details. **c** Correlation between RBD AUC, spike AUC, and Abbott index values (Pearson's $r = 0.93$, 0.95, and 0.96, respectively).

between symptom onset and seroconversion[10–14]. In addition, the temporal dynamics of seropositivity in our study population mirrors recent viral testing-based findings that infection in Seattle area children was rare before March 2020[22], but increased markedly in March and April[23].

We also examined how many seropositive children had received RT-PCR viral tests for COVID-19. This is an important question, since children are underrepresented in viral-testing case counts[3–5] and household-contact studies[24,25], and other studies[26–28] differ on whether children are less susceptible to infection than adults. Over a third of children in our study had received at least one viral test (Table 1; note that administration of a viral test does not imply that a child was suspected of having COVID-19, since tests were routinely administered before hospital admission or procedures such as surgery). Of the eight seropositive children, only two had tested positive for the virus (Table 1, Supplementary Table 1). Two other seropositive children had tested negative for the virus in routine screening prior to surgery. The remaining seropositive children never received a viral test. In addition, one child with a

positive viral test was seronegative; this is the child whose serum was collected <1 week post symptom onset (Fig. 2b, c), prior to when seroconversion is expected to occur.

A detailed chart review revealed that only the two seropositive children with a RT-PCR-confirmed viral infection had documented COVID-19 symptoms (Supplementary Table 1). One additional seropositive child, who presented at the hospital for an allergic reaction, had previous household exposure to the virus, but had not been tested because she did not develop symptoms. The five other seropositive children were at the hospital for reasons unrelated to respiratory illness (e.g., surgery and the underlying medical conditions) and had no documented exposures (Supplementary Table 1).

Other demographic and clinical data had limited noteworthy associations with seropositivity. There were more seropositive males than females, but the difference was not statistically significant ($P = 0.29$, Fisher's exact test, Table 1). Note that studies in China found no major sex differences in pediatric cases[6,8], but a recent report found more pronounced male sex

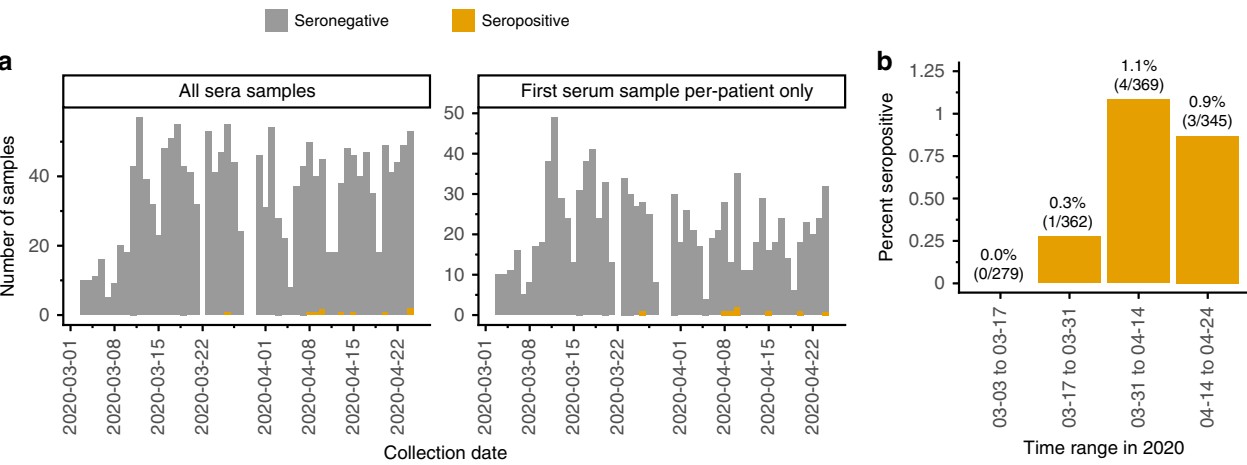

**Fig. 2 Frequency of seropositive samples over time. a** Total and seropositive samples collected each day, with stacked bars showing seropositive samples in orange and seronegative ones in gray. The left panel shows all samples, while the right panel shows only the first sample from each patient. **b** Percentage of tested patients with at least one seropositive sample during each 2-week period.

### Table 1 Cohort demographics.

|  |  | All children (n = 1076) | Seropositive children (n = 10) |
|---|---|---|---|
| Age (years) | 0–4 | 192 | 4 |
|  | 5–9 | 214 | 1 |
|  | 10–14 | 301 | 2 |
|  | ≥15 | 369 | 3 |
| Sex | F | 535 | 4 |
|  | M | 541 | 6 |
| RT-PCR viral-testing status | Positive | 3 | 2 |
|  | Negative | 389 | 3 |
|  | Not tested | 684 | 5 |
| Admit type | Outpatient | 653 | 4 |
|  | Inpatient | 306 | 4 |
|  | Emergency | 101 | 2 |
|  | Day surgery | 16 | 0 |

If a child had multiple samples, age and admit were determined based on the child's first visit. For viral-testing status, a child was classified as positive if they had a positive viral test at any visit.

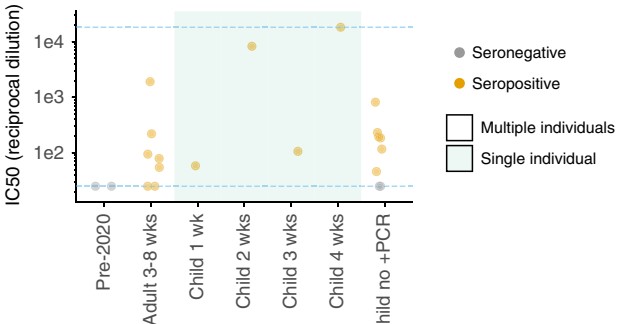

**Fig. 3 Neutralizing activity of sera against spike-pseudotyped lentiviral particles.** The y axis is reciprocal dilution of serum that inhibits infection by 50% (IC$_{50}$). Dashed blue lines are the limits of dilution series; points at those limits are lower or upper bounds. The child sera shown here are from the same individuals as in Fig. 1b; the two seronegative "child no +PCR" samples are the two ELISA+, Abbott CMIA− samples. Shaded categories indicate single individuals, as in Fig. 1b. Full curves are in Supplementary Fig. 1.

skewing in SARS-CoV-2 cases among pediatric oncology patients in New York[29]. The seropositive children spanned all ages from 0 to 4 years to ≥15 years, and were admitted to the hospital for a variety of reasons (Table 1, Supplementary Table 1).

**Neutralizing activity of pediatric seropositive samples.** Finally, we measured the neutralizing activity of sera from seropositive children using lentiviral particles pseudotyped with spike[30]. Sera from all eight seropositive children neutralized the virus at a >1:25 dilution (Fig. 3, Supplementary Fig. 1), including children that did not have documented symptoms. Neutralization correlated with RBD and spike binding as measured in the ELISAs (Supplementary Fig. 2). Two children had very high neutralizing activity, including one with a potency that exceeded the limit of our assay (>1:18,000 dilution). In comparison, the aggregated results of four studies of adults reported only 2 out of 263 individuals who had neutralizing activity >1:10,000 at 2–5 weeks post symptom onset using lentiviral pseudotype assays similar to our own[10,31–33]. Our sample size is too small to draw conclusions about neutralizing immunity in children versus adults, but this is an interesting area for further study, given that children develop stronger or quicker neutralizing responses to some other viruses[34–38].

In this study, we used serological assays to retrospectively identify SARS-CoV-2 infections in children early in the Seattle outbreak. Although our study used sera from children seeking medical care, and therefore does not represent an unbiased population survey, it nonetheless represents the first large-scale SARS-CoV-2 serological survey of children. Because we lack serological data from a comparable adult population in Seattle, our results are not sufficient to draw strong conclusions about the relative prevalence of SARS-CoV-2 infection of children versus adults in Seattle. However, as noted above, the frequency of seropositive samples in our study population is roughly similar to estimates of cumulative all-age incidence in Seattle based on testing and mortality data[20,21]. Comparing across age groups remains an important area for further investigation, as other very recent serosurveys that included both children and adults have reached differing conclusions about whether prevalence differs[39,40]. In any case, our work shows how serological assays can identify pediatric infections missed by the symptom-based administration of viral tests, as most seropositive children in our study had never tested positive for the virus. However, the overall frequency of seropositivity was low (≈1%) even in April, suggesting that while

infections of children are often missed by viral testing, perhaps due to the lack of symptoms, only a small fraction of children in Seattle had been infected by SARS-CoV-2 as of April 2020.

## Methods

**Study participants**. Residual sera samples at Seattle Children's Hospital were collected starting March 3, 2020. The adequate volume of sera remaining after other lab tests were conducted was the main sample-selection criterion, which inherently reduces the relative number of samples from infants who have smaller blood-draw volumes. Adult samples (Fig. 1b, c) were residual plasma collected from RT-PCR-confirmed inpatient cases from the Seattle area at the University of Washington, or from RT-PCR-confirmed outpatient individuals enrolled in a prospective cohort study. The sample collection and this study were approved by the Institutional Review Boards of Seattle Children's Hospital and the University of Washington. This study was granted a waiver of consent since it used residual clinical samples and existing clinical data.

Additional RT-PCR-confirmed COVID-19 pediatric cases were actively recruited to enroll in another approved study at Seattle Children's Hospital during this study period. These actively enrolled children were omitted from this seroprevalence study, which consisted of the residual serum sample pool at Seattle Children's Hospital and its clinics. However, pediatric samples from 1 to 4 weeks post symptom-onset facets in Figs. 1b, c and 3 are from these actively enrolled children, and were included in these plots to illustrate the sensitivity of our serological assays—however, these two children are not included in the estimates of seroprevalence in Fig. 2 or Table 1. In addition, in Fig. 1b, c, samples in the <1-week facet (two samples from the same individual), the 2-week facet (two samples from another individual), and the 3-week facet (a single sample from another individual) were RT-PCR-confirmed cases that were not recruited to Seattle Children's Hospital and therefore are included in the seroprevalence estimates.

**Serological assays**. We initially screened all sera at a 1:50 serum dilution for IgG binding to RBD. This was performed in four temporally grouped batches as samples became available (Fig. 1a). All sera were heat-inactivated at 56 °C for 1 h. Ninety-six-well Immunlon 2HB plates (Thermo Fisher, 3455) were coated with 2 µg/mL of His-tagged RBD in phosphate-buffered saline (PBS) overnight at 4 °C. The RBD antigen was produced in mammalian cells and purified as previously described[15,16,41]. The next day, the plates were washed three times with PBS containing 0.1% Tween 20 (PBS-T) using an automated plate washer (Tecan HydroFlex) and blocked for 1–2 h at room temperature with PBS-T containing 3% nonfat dry milk. Sera were diluted 1:50 in PBS-T containing 1% nonfat dry milk. The block was thrown off, and 100 µL of diluted sera were transferred to the ELISA plate in a setup as previously described[16]. Each plate also contained two positive-control wells (CR3022[18,19], an anti-SARS-CoV-1 monoclonal antibody that reacts to the SARS-CoV-2 RBD, at 0.5 µg/mL) and two negative-control wells (pooled human sera taken from 2017 to 2018 (Gemini Biosciences, 100–110, lot H86W03J, pooled from 75 donors)). CR3022 was expressed in Expi293F cells and purified by protein A and size-exclusion chromatography using established methods. After a 2-h incubation at room temperature, the plates were washed with PBS-T thrice. Goat anti-human IgG-Fc horseradish peroxidase (HRP)-conjugated antibody (Bethyl Labs, A80-104P) was diluted 1:3000 in PBS-T containing 1% milk, and 50 µL was added to each well. After 1 h at room temperature, the plates were washed thrice with PBS-T, and 100 µL of TMB/E HRP substrate (Millipore Sigma, ES001) was added to each well. After 5 min, 100 µL of 1 N HCl was added, and OD450 was read immediately on a Tecan infinite M1000Pro plate reader. Samples were considered potential positive hits in the screen if their reading exceeded the average of all of the 2017–2018 negative-control readings by >5 standard deviations, computing this threshold separately for each screening batch (Fig. 1a).

Follow-up ELISAs were performed on all potential positive hits from the screening assay, plus a subset of samples that were negative in the initial screen ($n = 30$ for the RBD titration, and $n = 32$ for the spike titrations; all samples that proved seropositive by our criteria were positive in the initial screen). These follow-up ELISAs were performed as for the screening step described above, with the following differences: all sera were run at five 3-fold dilutions, starting at 1:25. Each plate contained a negative-control dilution series (pooled human sera taken from 2017 to 2018), and a CR3022 positive-control dilution series starting at a concentration of 1 µg/mL. Trimeric, prefusion-stabilized spike was produced as previously described[41] with the following minor changes: the protein was produced using expiHEK293F cells transfected transiently with PEI, and the cultures were grown at 33 °C for 3 days prior to downstream talon-batch purification. The additional control samples tested during follow-up ELISAs were pooled human sera from 2008 to 2015 (Gemini Biosciences, 100–110, lot H87W00K, pooled from 156 donors), and 12 de-identified banked serum samples collected between 1986 and 1992 (Bloodworks Northwest).

The Abbott SARS-CoV-2 IgG CMIA was run on all samples that were run in follow-up ELISAs that still had adequate sample volume remaining ($n = 124$, with 10 samples omitted due to volume [8 RBD screen positive and 2 RBD screen negative, all seronegative by follow-up ELISAs and classified as seronegative overall]). This assay, which detects IgG antibodies to SARS-CoV-2 nucleocapsid protein, was run on the Abbott Architect instrument according to the manufacturer's instructions. We used the manufacturer's recommended positivity index value cutoff of 1.40.

**AUC analysis**. For the follow-up ELISAs performed at serial dilutions, the AUC represents the area under the titration curve after putting the serial dilutions on a log scale (as plotted in Fig. 1b). Readings for the 2017–2018 pooled sera that were run on each plate were first averaged and treated as a single sample. This, along with the 12 banked pre-2020 sera samples and the additional 2008–2015 pooled sera, were treated as 14 negative controls and used to determine a cutoff in each assay (average of all negative controls plus 5 standard deviations).

**Neutralization assays**. SARS-CoV-2 spike-pseudotyped lentivirus-neutralization assays were performed as previously described[30], with the following slight modifications. Infections were carried out in poly-L-lysine- (P4707, Millipore Sigma, Burlington, MA, USA) coated black-walled, clear-bottom plates (655090, Greiner Bio-One, Kremsmünster, Austria), and luciferase activity was measured in these plates without transferring to opaque-bottom plates. Sera were diluted 3-fold 7 times starting at a 1:25 dilution, and luciferase activity was measured at 52 h post infection. Target cells were HEK-293T cells transduced to express hACE2[30] (BEI Resources, NR-52511). Samples were run in duplicate, and each plate included two no-serum controls. Fraction infectivity was calculated by normalizing the luciferase reading for each sample by the average of the two no-serum control wells in the same row. Neutralization curves were plotted using the neutcurve Python package (https://jbloomlab.github.io/neutcurve/, 0.3.1). This package fits a three-parameter Hill curve, with the top baseline fixed to one and the bottom baseline fixed to zero. The neutralization curves are shown in Supplementary Fig. 1.

**Period seroprevalence analysis**. Period seroprevalence was calculated over 2-week periods at the individual level. We calculated the percentage of all tested patients that had a seropositive sample during each time period. Individuals were counted a single time even if multiple samples from a single time period were tested. If an individual contributed samples to multiple time periods ($n = 279$), they were counted for each time period.

**Reporting summary**. Further information on research design is available in the Nature Research Reporting Summary linked to this article.

## Data availability

Full raw data for all serological assays, as well as much demographic and viral-testing data that can be provided without compromising sample and patient de-identification, are available in Supplementary Data files 1–4.

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

## Acknowledgements

We are grateful to all study participants for contributing samples. We also thank Abigail Powell, Peter Rupert, and Roland Strong for sharing antigens, Elizabeth Ahearn for laboratory management at Seattle Children's Hospital, and Mike Famulare and Alex Greninger for helpful comments on this paper. This work was supported by the NIAID/NIH (R01AI141707 and R01AI140891 to J.D.B., HHSN272201700059C to D.V., and F30AI149928 to K.H.D.C.), the NIMGS/NIH (R01GM120553 to D.V.), the Bill & Melinda Gates Foundation (OPP1156262 to N.P.K.), a Pew Biomedical Scholars Award (to D.V.), and Burroughs Wellcome Investigators in the Pathogenesis of Infectious Diseases awards (to D.V. and J.D.B.). J.D.B. is an investigator of the Howard Hughes Medical Institute. Work in the Krammer laboratory was partially supported by the NIAID Centers of Excellence for Influenza Research and Surveillance (CEIRS) contract HHSN272201400008C, Collaborative Influenza Vaccine Innovation Centers (CIVIC) contract 75N93019C00051, and philanthropic donations.

## Author contributions

Conceptualization: A.S.D., J.A.E., and J.D.B.; investigation: A.S.D., K.H.D.C., R.E., and J.A.D.; analysis: A.S.D., K.H.D.C., and J.D.B.; clinical management, sample, and data handling: A.A., S.S., K.L., C.R.W., X.Q., H.Y.C., and J.A.E.; resources and specialized reagents: A.C.W., C.R.W., F.A., N.P.K., D.V., F.K., and H.Y.C.; writing—original draft preparation, A.S.D and J.D.B.; writing—review and editing: all authors. All authors have read and agreed to the published version of the paper.

## Competing interests

H.Y.C. is a consultant for Merck and Glaxo Smith Kline and receives research funding from Sanofi Pasteur, outside of the submitted work. N.P.K. is a co-founder, shareholder, and chair of the scientific advisory board of Icosavax, Inc. Mount Sinai has licensed serological assays to commercial entities and has filed for patent protection for serological assays. J.A.E. is a consultant for Sanofi Pasteur and Meissa Vaccines. The remaining authors declare no competing interests.
