## [Peer Review File · Nature Communications]

REVIEWERS' COMMENTS:

Reviewer #1 (Remarks to the Author):

This is the second round of reviews for this paper (as far as I know). I have gone through this updated manuscript and compared with the first version. The authors have done a superb job addressing every point from the previous version and I feel this is now in very good shape for publication in Nature Communications. The paper remains timely and important and I encourage its rapid publication and dissemination, particularly given the current intense focus in children and COVID-19.

Terrific work to the authors.

Below is the review and review from *Nature Communications* in blue. Our responses to the review is in line in black.

REVIEWERS' COMMENTS:

Reviewer #1 (Remarks to the Author):

This is the second round of reviews for this paper (as far as I know). I have gone through this updated manuscript and compared with the first version. The authors have done a superb job addressing every point from the previous version and I feel this is now in very good shape for publication in *Nature Communications*. The paper remains timely and important and I encourage its rapid publication and dissemination, particularly given the current intense focus in children and COVID-19.

Terrific work to the authors.

We thank the reviewer for their comments here and during the first review process. We agree with the reviewer that rapid publication and dissemination of this work is critical and timely.